# The Clinical and Molecular Profile of Lung Cancer Patients Harboring the *TP53* R337H Germline Variant in a Brazilian Cancer Center: The Possible Mechanism of Carcinogenesis

**DOI:** 10.3390/ijms242015035

**Published:** 2023-10-10

**Authors:** Carlos D. H. Lopes, Fernanda F. Antonacio, Priscila M. G. Moraes, Paula F. Asprino, Pedro A. F. Galante, Denis L. Jardim, Mariana P. de Macedo, Renata L. Sandoval, Artur Katz, Gilberto de Castro, Maria Isabel Achatz

**Affiliations:** 1Hospital Sirio Libanes, São Paulo 01308-050, Brazil; ffantonacio@gmail.com (F.F.A.); primed69@gmail.com (P.M.G.M.); pasprino@mochsl.org.br (P.F.A.); pgalante@mochsl.org.br (P.A.F.G.); jardimde@gmail.com (D.L.J.); maripetaccia@gmail.com (M.P.d.M.); rsandoval.med@gmail.com (R.L.S.); arturkatz@gmail.com (A.K.); gcastrojr@terra.com.br (G.d.C.J.); 2Oncoclínicas, São Paulo 04543-906, Brazil

**Keywords:** lung cancer, *TP53* R337H variant, Li-Fraumeni syndrome, lung adenocarcinoma, molecular profiling, particulate matter, environmental pollution

## Abstract

In southern and southeastern Brazil, the *TP53* founder variant c.1010G>A (R337H) has been previously documented with a prevalence of 0.3% within the general population and linked to a heightened incidence of lung adenocarcinomas (LUADs). In the present investigation, we cover clinical and molecular characterizations of lung cancer patients from the Brazilian Li-Fraumeni Syndrome Study (BLISS) database. Among the 175 diagnosed malignant neoplasms, 28 (16%) were classified as LUADs, predominantly occurring in females (68%), aged above 50 years, and never-smokers (78.6%). Significantly, LUADs manifested as the initial clinical presentation of Li-Fraumeni Syndrome in 78.6% of cases. Molecular profiling was available for 20 patients, with 14 (70%) revealing *EGFR* family alterations. In total, 23 alterations in cancer driver genes were identified, comprising 7 actionable mutations and 4 linked to resistance against systemic treatments. In conclusion, the carriers of *TP53* R337H demonstrate a predisposition to LUAD development. Furthermore, our results indicate that environmental pollution potentially impacts the carcinogenesis of lung tumors in the carriers of *TP53* R337H.

## 1. Introduction

In 2022, the American Cancer Society estimated a total of 236,740 new cases of lung cancer (LC), resulting in 130,180 deaths, accounting for about 25% of all cancer-related deaths [1]. In Brazil, 32,560 new cases were predicted by 2023, and it is the fourth most common malignancy by incidence [2]. Although most cases are sporadic, some cases are caused by external agents such as tobacco smoking, air pollution, and other environmental carcinogens [3]. In the past few years, there has been a growing concern regarding outdoor air pollution as a carcinogen for LC, and its effect may be measured by particulate matter (PM) [4]. The PM_2.5_ (fine particulate matter with <2.5 µm of diameter) may enter the terminal bronchioles and alveoli, triggering local inflammatory processes [5]. Incidence and mortality from LC with PM_2.5_ levels have been demonstrated in North America, Asia, and Europe regardless of cigarette smoke exposure [6].

Germline pathogenic variants in the genes involved in carcinogen clearance, DNA repair, and cell cycle control may confer higher tumor-specific risk [7,8,9]. Recently, a retrospective analysis of 7788 lung cancer patients who performed exome sequencing revealed the presence of germline pathogenic variant in 15% of cases, a higher prevalence than previously described for *BRCA2*, *CHEK2*, *ATM*, *TP53*, *BRCA1*, and *EGFR* [10,11]. Notably, some of these genes are potentially targetable [11].

Li-Fraumeni syndrome (LFS) is a rare heritable condition resulting from germline pathogenic variants in the tumor suppressor gene *TP53*. Carriers are at a high lifetime risk of developing a large spectrum of cancers, including early-onset breast, sarcoma, central nervous system, adrenocortical, and lung malignancies [12]. According to the IARC database, 3% of individuals harboring *TP53* mutations develop LC, which represents the eighth tumor in incidence in the context of LFS [12].

In Brazil, the c.1010G>A p.Arg337His (R337H) variant in *TP53* is present in 0.3% of the southern and southeastern population due to a founder effect [13]. Unlike the vast majority of *TP53* pathogenic variants, which occur in the DNA-binding domain (exons 5–8), R337H occurs at the oligomerization domain in exon 10 and has a milder penetrance than other variants. Previous studies in nonselected lung adenocarcinoma (LUAD) populations from Brazil found an enrichment of the R337H variant, with a prevalence ranging from 2.3% to 5.4% and a statistical correlation to *HER* family alterations [14,15,16]. However, limited clinical and demographic information did not permit further conclusions.

The present study aims to describe the clinical and molecular features of R337H carriers diagnosed with LC in Brazil. Furthermore, we postulate some possible mechanisms involved in LC carcinogenesis in the context of R337H LFS.

## 2. Results

### 2.1. Demographic, Clinical Features, and Molecular Profile

From a total of 370 individuals with data in the BLISS database, 175 (47.3%) were diagnosed with cancer (Figure 1A), 28 of whom (16%) had lung cancer (LC) (Figure 1B). Demographic and clinical features are presented in Table 1. The mean age at LC diagnosis was 54.6 years old (ranging from 22 to 72), with a predominance of women (n = 19, 68%; vs. men n = 9, 32%). Adenocarcinoma was the sole histology found, and 78.6% (n = 22) were nonsmokers. Interestingly, LC was the first tumor diagnosed in 78.6% (n = 22), with 15 cases (53.8%) showing no evidence of secondary neoplasms. However, 42.9% (n = 12) had a second tumor. At the time of diagnosis, staging was categorized as I–II (n = 12; 42.9%), III (n = 5; 17.9%), and IV (n = 10; 35.7%). Tumoral driver assessment was performed in 71.4% (n = 20) of the cases, using next-generation sequencing in 16 (employing different platforms evaluating 180–505 genes), and real-time PCR in 4 patients (specifically for *EGFR* mutations and *ALK* fusions). Among these 20 cases, 70% (n = 14) had *EGFR* alterations located in exon 19 (n = 3), L858R (n = 7), exon 18 (n = 1), exon 20 (n = 2), and *HER2* mutation (n = 1), as depicted in Table 2. Four patients (#1, #4, #8, and #17) were only tested for *EGFR* and *ALK* alterations. No mutations were described in those cases (Figure 2). The somatic tumoral genomic profile is summarized in Figure 2, with a total of 23 driver alterations identified. Seven were actionable (*EGFR*, *HER2*, *NF1*, *BRAF*, *VHL*, *KRAS*, and *MSH2*), and four were related to resistance to systemic treatments (*CDKN2a/b*, *EGFR* amplification, *RB1*, and *STK11*).

### 2.2. Clinical Outcomes

Among those patients in early and locally advanced stages (stages I–III), 16 underwent surgery and attained a median disease-free survival (time from tumor resection to recurrence or death for any cause) of 106.9 months (range: 0.03–229.8 months). Among them, 68.8% (n = 11) exhibited no evidence of disease, while 32.1% (n = 5) experienced recurrences (depicted in Figure 3a). At the latest follow-up, the median overall survival (time from initial cancer diagnosis to death from any cause) was not reached since none of the patients died (Figure 3b).

For individuals with advanced disease and *EGFR* mutations (n = 11), the median progression-free survival was 19.6 and 17.4 months for the first-generation (erlotinib and gefitinib) and third-generation (osimertinib) *EGFR* inhibitors, respectively, as illustrated in Figure 4a,b. In this context, the median overall survival was not reached, as shown in Figure 4c,d. Notably, one death occurred during osimertinib treatment, coinciding with the emergence of *BRAF* amplification.

In a patient harboring a somatic *EGFR* exon 20 insertion (D770_N771insY), an experimental *EGFR* oral inhibitor targeting exon 20 alterations was administered as the sixth line of treatment, resulting in partial response (for lesions in the lungs and brain) and symptom control for seven months (Figure 5a), followed by multifocal progression. Another patient with *HER2* exon 20 insertion and liver metastasis received four lines of therapy (afatinib, carboplatin–paclitaxel, pertuzumab–trastuzumab–docetaxel, and trastuzumab–entansine, in that order) for a cumulative duration of 25 months until disease progression to the central nervous system (Figure 5b).

In the case of four metastatic patients who lacked *EGFR* family activation mutations and were subjected to chemoimmunotherapy, it should be noted that three of them experienced an early loss of follow-up, occurring within a period of less than four months, rendering them ineligible for inclusion in the clinical outcomes analysis. The remaining patient died after 42 months of systemic treatment.

### 2.3. PM_2.5_ Measurements

All cases were categorized across 17 cities in six Brazilian States (as outlined in Table 3), with the majority (n = 8) situated in the state of São Paulo, followed by Minas Gerais (n = 3), Paraná (n = 3), Federal District (n = 1), Goiás (n = 1), and Santa Catarina (n = 1). These cities correspond to three distinct regions within Brazil, namely Southeast, Central East, and South. All patients were residing in these cities at the time of their lung cancer (LC) diagnosis.

In seven of these cities (four in São Paulo, one in the Federal District, one in Paraná, and one in the State of Minas Gerais), accounting for a total of 18 cases, we searched for the official government environmental pollution data in the form of PM_2.5_ measurements from ground-based monitoring stations (Table 3) and available at the Air Quality Historical Data Platform website. Available online: https://aqicn.org/data-platform/register/ (accessed on 16 January 2023). These measurements ranged from 63.9 (in the city of São Paulo) to 30 (in the city of Brasília) and surpassed the PM_2.5_ levels specified as indicative of poor air quality by the European Ambient Air Quality Directives (as depicted by the horizontal red dotted line in Figure 6b) [17].

We visually depicted the distribution of LC cases on a map of Brazil, accentuating regions with elevated PM_2.5_ concentrations based on satellite imagery sourced from the IQAIR website [Available online: https://www.iqair.com (accessed on 16 January 2023)] (Figure 6a), which coincided with the locations of the patients’ cities.

## 3. Discussion

As previously described, the *TP53* R337H Brazilian germline founder mutation had a general population incidence of 0.3% in southeastern and southern regions, with less penetrance besides a differential spectrum of tumors [13,18,19,20]. In a retrospective pan-cancer analysis in Brazil, Sandoval and collaborators described a 2% incidence of this variant, highlighting the high prevalence in that territory [21]. Characteristically, this variant is predominantly described as germline, with few cases reported as somatic [22,23], and some specialists recommend germinal evaluation when this alteration is found in somatic assays.

Some studies conducted in Brazil have described an enrichment of lung adenocarcinomas (LUADs) in R337H carriers. Starting from R337H carriers, Barbosa et al. found an incidence of LUADs of 5.4%, with most of these having activating mutations in *EGFR* (eight of nine patients) [15]. Another retrospective analysis of 513 patients with LC (83.8% LUADs) described *TP53* mutation in 53.6% of the cases using somatic NGS assay, 2.3% of these corresponding to the R337H variant (12 patients), which had a statistical association to *EGFR* and *HER2* alterations [16]. However, a limited number of patients or clinical information did not allow for further conclusions in these studies.

Here, we describe the largest cohort of patients with LC in a *TP53* R337H setting (28 patients), representing 16% of the 175 cases with diagnosed tumors from the BLISS database. In line with other series [15,16], we found a predominance of women (68%), with the age of above 50 years old at diagnosis (mean of 54.6); all cases were adenocarcinoma, in nonsmokers (78.6%), and with alterations in the *ERBB* family (70% of the tested cases; 14 of 20 patients) concentrated in exons 18 to 20 (to the *EGFR*), and one case with *HER 2* exon 20 insertion. Interestingly, LUAD was the first tumor diagnosed in 78.6% of cases, which was not previously described. These findings reinforce the possibility of interactions between the *TP53* R337H variant and the *ERBB* family in LUAD carcinogenesis; however, more studies are needed to elucidate the possible mechanism of these findings. Most diagnoses were in the context of an initially resectable disease, which may be due to the profile of patients seen in a private service, where the search for medical assistance occurs earlier, or due to incidental findings in imaging tests. Because of the small number of cases, we did not perform statistical analysis, and the clinical outcomes were only described. For exposure, none of the patients with localized disease who underwent surgery died (n = 15), and they had a median disease-free survival of 106.9 months. In patients with metastatic disease and activating *EGFR* mutations treated with *EGFR* inhibitors, there was no numerical difference in progression-free survival between patients on first-generation (erlotinib and gefitinib) or third-generation (osimertinib) *EGFR* inhibitors. 

A previous analysis of a large patient population found a positive association between air pollutants and tumor incidence, such as brain tumors [23] and lung cancer [24], with emphasis on chronic high levels of PM_2.5_ in µg/m^3^ [6]. Experimentally, the exposition of human bronchial epithelial cell lines to airborne particulate matter (PM) results in an inflammatory response as dose-dependent reduced cell viability and increased reactive oxygen species production [25]. An elegant study published by Hill et al. proved that exposing airway cells that already harbor driver mutations, such as *EGFR* and *KRAS*, to PM_2.5_ would induce a stem cell phenotype that, ultimately, results in the formation of adenocarcinomas. In this carcinogenesis model, the production of IL-1ß by infiltrating macrophages would be a pivotal step in the process since the use of this cytokine antagonist was able to inhibit tumor formation [26].

According to this, we found high PM_2.5_ levels in the cities of the patients in our study (Table 3 and Figure 6b), with mean daily levels above the threshold designated as poor by the European Air Quality Index, and the LUAD diagnosis in *TP53* R337H context concentrate in areas of more pollution levels in Brazilian map (Figure 6a). In addition, in all cases (n = 16) with somatic NGS assessment, a driver mutation was found (besides the *TP53* R337H), which could reinforce the hypothesis that pollution exposure could promote cancer development in already initiated bronchial cells [27]. Finally, some *TP53* variants, but not the normal gene, could enhance the IL-1ß activity by suppressing the production of secreted interleukin-1 receptor antagonists in cancer cell and xenograft models [28]. Future studies are needed to elucidate the possible mechanisms of interaction between the *TP53* R337H variant and IL-1β signaling in the context of exposure to high levels of PM_2.5_ in the carcinogenesis of lung adenocarcinoma.

The lack of somatic NGS results is a significant limitation of this study. Furthermore, the absence of data regarding the duration of residence in cities with elevated levels of PM_2.5_ prior to the tumor diagnosis, and the lack of a control group carrying the R337H variant without the occurrence of lung cancer development, render the specific correlation between air pollution and LUAD carcinogenesis a matter of mere speculation. Further studies are needed to address this inquiry. However, we assembled the largest cohort of patients with LUADs in the LFS R337H context, describing the clinical characteristics, clinical outcomes, and tumor somatic molecular profile in more detail, in addition to formulating hypotheses for the role of *TP53* R337H in the carcinogenesis of these tumors.

## 4. Materials and Methods

### 4.1. Demographic, Clinical, and Genomic Information

We conducted a single-center retrospective analysis of 370 individuals enrolled in the Brazilian Li-Fraumeni Syndrome Study (BLISS) registry at Sírio-Libanes Hospital, São Paulo City, Brazil [16]. All individuals harbored a *TP53* R337H germline pathogenic variant, which was confirmed through commercial germline testing methods, including Sanger sequencing, multiplex ligation-dependent probe amplification (MLPA), or next-generation sequencing platforms, all of which are covered by private health insurance in our country. Demographic and clinical data, including age at diagnosis, gender, smoking history, initial tumor diagnosis, history of other neoplasms, lung cancer histology, and staging at diagnosis, were collected for patients treated between 2010 and 2022, and they are presented in tables and diagrams based on variable characteristics. Furthermore, we documented the presence and methodology of somatic molecular tumor assessments in a table and diagram, highlighting the molecular profiles.

### 4.2. Clinical Outcomes

Using the Kaplan–Meier method, we established the medians of disease-free survival (time from definitive treatment to the recurrence of tumor or death due to any cause), progression-free survival (time from the beginning of systemic therapy to documented progression or death due to any cause), and overall survival (time from diagnosis to death due to any reason). Data were censored after the last present evaluation in the reference center for patients who did not have events. The Kaplan–Meier curves were generated by using GraphPad Prism version 8.

### 4.3. Measurement of Pollution Level

To investigate the potential correlation between environmental pollution and the incidence of lung cancer, we determined the daily mean PM_2.5_ levels, measured in µg/m^3^, using the data of the respective cities of the patients. These measurements were obtained using ground-level monitoring devices, and the data sources were the official government environmental agencies of each state, which can be accessed at the Air Quality Historical Data Platform website. Available online: https://aqicn.org/data-platform/register/ (accessed on 16 January 2023). The PM_2.5_ values were then depicted using box plots and compared against the air quality standards set forth by the European Union [18]. Logarithmic transformations were applied to these data to enhance graphical representation. 

## 5. Conclusions

Individuals carrying the Brazilian founder mutation *TP53* R337H exhibit a heightened incidence of lung cancer compared with more prevalent global variants. Within the *TP53* R337H context, lung cancer may manifest as the primary indication of Li-Fraumeni syndrome, predominantly diagnosed in never-smoking women over the age of 50. All the observed cases were adenocarcinomas, with the majority featuring driver mutations highlighted by actionable alterations in the *ERBB* family. Descriptively, it appears that these tumors may yield benefits from systemic therapies in advanced settings. As a speculative hypothesis, our data indicate a potential correlation between elevated environmental pollution levels and the diagnosis of lung cancer in this scenario, emphasizing the need for more rigorous studies to elucidate this possible association.

## Figures and Tables

**Figure 1 ijms-24-15035-f001:**
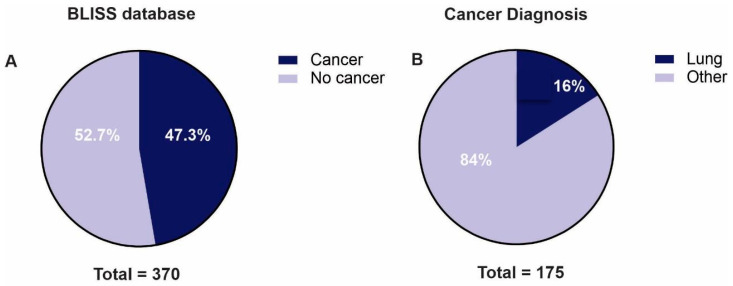
Patients included in the BLISS database: (**A**) proportion of patients with cancer diagnosis; (**B**) proportion of lung cancer from all tumors.

**Figure 2 ijms-24-15035-f002:**
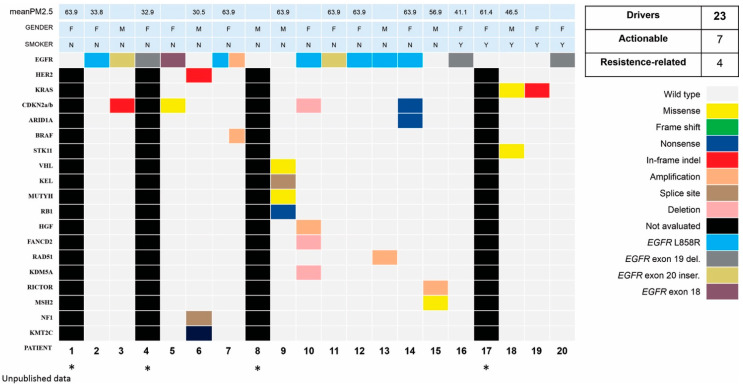
Genomic landscape of lung adenocarcinomas. Genomic profile of the 20 patients submitted to tumoral multigenic next-generation sequencing (NGS) platforms or real-time PCR (rtPCR) for *EGFR* and *ALK*. NGS was performed in 16 patients and rtPCR was performed in 4 (indicated by *). Most cases indicated mutations in driver genes. A total of 23 alterations, comprising 7 actionable and 4 related to systemic treatment resistance, were described. In horizontal lines, individual characteristics of the patients are listed, such as average levels of exposure to PM_2.5_, gender, smoking, and evaluated genes. Each column indicates an individual patient. On the right, the genetic alterations are described and illustrated with colored squares. Abbreviations: del—deletion; indel—insertion or deletion; inser—insertion; PM_2.5_—particulate matter < 2.5mm of diameter.

**Figure 3 ijms-24-15035-f003:**
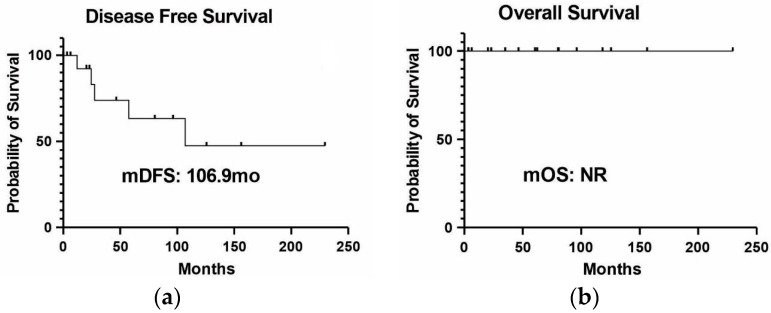
Clinical outcomes of 16 patients with early disease submitted to surgery: (**a**) median disease-free survival of the sixteen patients assessed; (**b**) overall survival of these patients. None of them had died due to LC. Abbreviation: NR—not reached.

**Figure 4 ijms-24-15035-f004:**
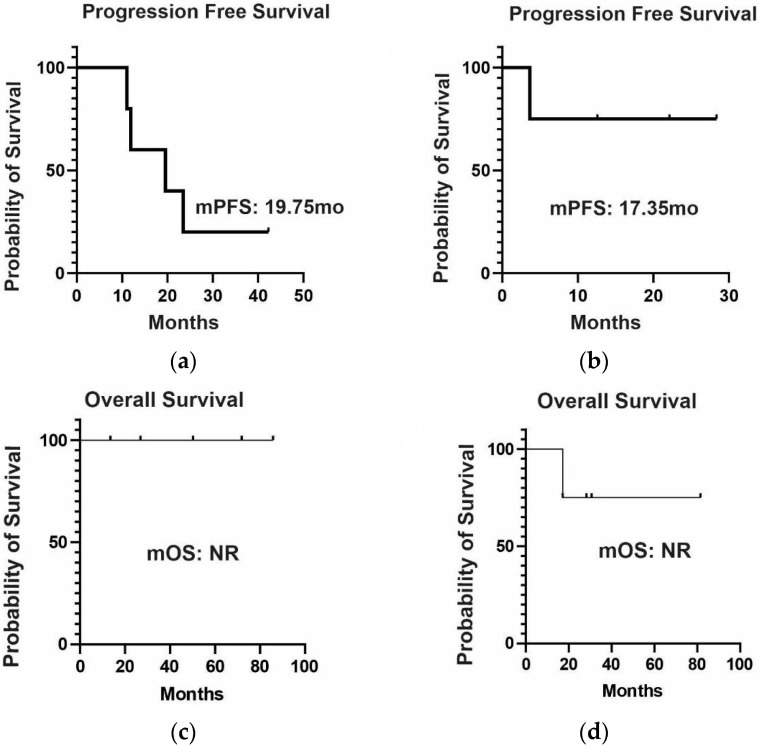
Clinical outcomes in 11 patients with advanced disease treated with *EGFR* inhibitors: (**a**) median progression-free survival on first-generation *EGFR* inhibitor treatment (n = 6; median follow-up of 15.7 months); (**b**) median progression-free survival on third-generation *EGFR* inhibitor treatment (n = 5; median of follow-up of 12 months); (**c**) median overall survival on first-generation *EGFR* inhibitor treatment (n = 6; median of follow-up of 50.3 months); (**d**) median overall survival on third-generation *EGFR* inhibitor treatment (n = 5; median of follow-up of 67.6 months).

**Figure 5 ijms-24-15035-f005:**
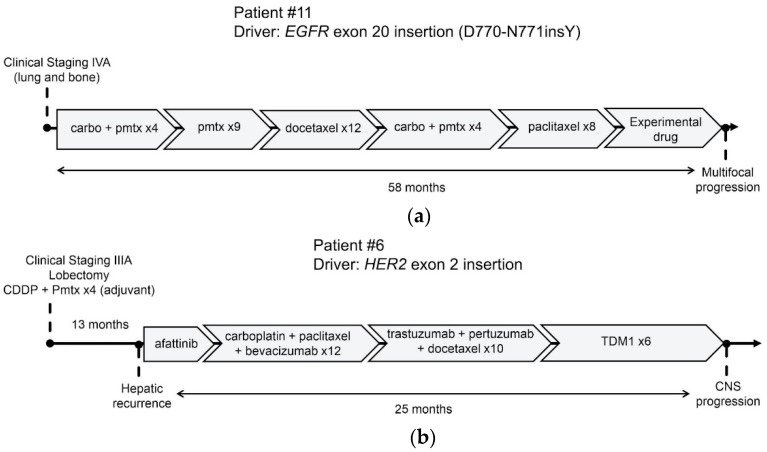
Clinical evolution on systemic treatments of patients #6 and #11: (**a**) a 41-year-old man with metastatic LC to bone and lungs with *EGFR* exon 20 insertion was submitted to six lines of treatment with clinical and radiologic benefit until multisystemic progression after seven months of an experimental *EGFR* oral inhibitor for exon 20 alterations; (**b**) a 33-year-old man with previous resected LC harboring, *HER* 2 exon 20 insertion, and submitted to four cycles of platinum doublet experienced a hepatic recurrence 13 months after. Then, he received three lines of systemic treatment, as represented above, with clinical and radiographic benefits for 25 months until CNS progression. Abbreviations: carbo—carboplatin; CDDP—cisplatin; pmtx—pemetrexed; TDM-1—trastuzumab–emtansine; x followed by a number indicates the number of treatment cycles.

**Figure 6 ijms-24-15035-f006:**
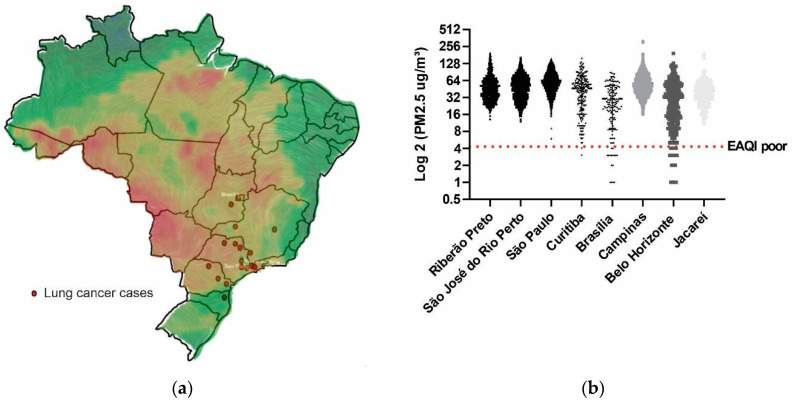
Distribution of cases in Brazil and average daily values of PM_2.5_: (**a**) location of patients diagnosed with LC according to the PM_2.5_ heat map obtained via satellite (the redder the colors, the higher the PM_2.5_ value); (**b**) daily mean values are represented in box plots of cities with cases and comparison with the value established by the European Society for the Environment as poor air quality (EAQI poor). We applied logarithmic scaling to the values for better graphical representation. Note that most values are above the levels designated as EAQI poor.

**Table 1 ijms-24-15035-t001:** Demographic and clinical of LC LFS R337H patients.

Clinical Variables	Overall Population n = 28 (%)
Mean age of diagnosis (range)	54.6 (22–72) years
Gender:	
Female	19 (68%)
Male	9 (32%)
Adenocarcinoma	28 (100%)
Smoking status:	
No	22 (78.6%)
Yes	5 (17.9%)
Unavailable	1 (3.6%)
LC as first tumor	22 (78.6%)
Secondary neoplasia:	
No	15 (53.8%)
Yes	12 (42.9%)
Unavailable	1 (3.6%)
Stage at diagnosis (AJCC 8th ed.) ^1^:	
I–II	12 (42.9%)
III (A/B)	5 (17.9%)
IV (A/B)	10 (35.7%)
Molecular assessment:	
No	8 (28.6%)
Yes	20 (71.4%)
Type of molecular assessment:	
NGS	16 (57.1%)
RT-PCR for *EGFR* and *ALK*	4 (14.3%)

^1^ Missing one patient data.

**Table 2 ijms-24-15035-t002:** ERRB family’s mutational profile of LC LFS R337H patients who performed some tumoral molecular assessment.

*ERBB* Family Status	Population n = 20 (%)
Mutated/Wild-type	14 (70%)/6 (30%)
*EGFR* exon 19 deletion	3 (21.4%)
*EGFR* exon 18 mutation	1 (7.1%)
*EGFR* exon 20 insertion	2 (14.3%)
*EGFR* L858R	7 (50%)
*EGFR HER* 2 insertion	1 (7.1%)

**Table 3 ijms-24-15035-t003:** Cities and states distribution of LC cases with levels, methods, and periods for measurement of PM_2.5_ levels.

City	Estate	Number of Cases	Ages of LC Diagnosis (Years)	Year of LC Diagnosis	Mean PM_2.5_ (µg/m^3^)	Method of Measurement	Period of Measurement
Ribeirão Preto	SP	2	57/22	2012/2003	51.7	meter on land	2018–2022
São Paulo	SP	8	53/72/40/47/58/55/49/70	2014/2019/2020/2013/2018/2015/2021	63.9	meter on land	2014–2022
Santana do Parnaíba	SP	1	72	2017	12.8	-	-
São José do Rio Preto	SP	1	52	2016	51.8	meter on land	2014–2022
Jacarei	SP	1	63	2020	42	meter on land	2018–2019
Sertãozinho	SP	1	57	2020	25.6	-	-
Campinas	SP	1	41	2022	56.9	meter on land	2015–2022
Mogi das Cruzes	SP	1	62	2021	19.5	-	-
Uberlândia	MG	1	59	2019	27.7	-	-
Monte Sião	MG	1	55	2021	26.9	-	-
Goiânia	GO	1	51	2015	33.8	-	-
Brasília	DF	1	33	2016	30.5	meter on land	April–September/22
Curitiba	PR	3	59/62/44	2017/2009/2015	46.5	meter on land	2021–2022
Belo Horizonte	MG	1	49	2009	32.2 (0–193)	meter on land	February–September/22
Ponta Grossa	PR	3	67/35/60	2014/2017/2017	18.1	-	-
Londrina	PR	1	57	2016	32.9	-	-
Blumenau	SC	1	66	2012	12.2	-	-

From left to right, the vertical columns display the associated cities and states, along with the patients’ ages and years of diagnosis. Subsequently, the mean daily PM_2.5_ values, methodologies, and the duration of measurement are presented.

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
