# Peer review of "The Clinical and Molecular Profile of Lung Cancer Patients Harboring the TP53 R337H Germline Variant in a Brazilian Cancer Center: The Possible Mechanism of Carcinogenesis"

_ijms, 2023, doi:10.3390/ijms242015035_

Round 1

Reviewer 1 Report

This paper by Lopes et al. describes the TP53 founder variant R337H in a Brazilian subpopulation. The present paper is in good standing, but my opinion is that it requires several changes and improvements before it can be considered for publication.

1. First and foremost, it would be better if Table 1 contained only clinical data. I would be pro for a new table regarding the mutational status. Data should be better organized in general, as for the reader to get a faster and more comprehensive view upon this re-organized data.

2. Regarding Figure 2, 'Not avaluated' should read 'Not evaluated'. I suggest changing the patterns for the EGFR mutations, as the current patterns seem misleading.

3. Line 104 - not only early stage but locally advanced as well?

4. Figure 4 legend should be amended, as the alphabetic numbering is off.

5. I would critically re-check figure 6b.

6. Lines 220-221 first and third generation of EGFR inhibitors

7. Another limitation that is worth mentioning - the variable time spent by the patients in this media (with increased PM2.5 values), as some may have spent only a limited amount of time compared to others.

8. I consider that the Methods section should be improved.

9. I would also like the authors to improve the concluding remarks.

10. A re-check of the English with slight improvements is expected.

A re-check of the English with slight improvements is expected.

Reviewer 2 Report

Please find my comments in the attachment.
